# *Pneumocystis jirovecii* Pneumonia Associated with COVID-19 in Patients with Interstitial Pneumonia

**DOI:** 10.3390/medicina58091151

**Published:** 2022-08-24

**Authors:** Tomoyuki Takahashi, Atsushi Saito, Koji Kuronuma, Hirotaka Nishikiori, Hirofumi Chiba

**Affiliations:** Department of Respiratory Medicine and Allergology, Sapporo Medical University School of Medicine, Sapporo 060-8543, Japan

**Keywords:** SARS-CoV-2, COVID-19, *Pneumocystis jirovecii* pneumonia, interstitial pneumonia, steroids, immunosuppressive drugs

## Abstract

Here, we report two cases of patients with interstitial pneumonia (IP) on steroids who developed *Pneumocystis jirovecii* pneumonia (PJP) following coronavirus disease 2019 (COVID-19) infection. Case 1: A 69-year-old man on 10 mg of prednisolone (PSL) daily for IP developed new pneumonia shortly after his COVID-19 infection improved and was diagnosed with PJP based on chest computed tomography (CT) findings and elevated serum β-D-glucan levels. Trimethoprim–sulfamethoxazole (TMP–SMZ) was administered, and the pneumonia resolved. Case 2: A 70-year-old woman taking 4 mg/day of PSL for IP and rheumatoid arthritis developed COVID-19 pneumonia, which resolved mildly, but her pneumonia flared up and was diagnosed as PJP based on CT findings, elevated β-D-glucan levels, and positive polymerase chain reaction for *P. jirovecii* DNA in the sputum. The autopsy revealed diffuse alveolar damage, increased collagen fiver and fibrotic foci, mucinous component accumulation, and the presence of a *P. jirovecii* cyst. In conclusion, steroids and immunosuppressive medications are well-known risk factors for PJP. Patients with IP who have been taking these drugs for a long time are frequently treated with additional steroids for COVID-19; thus, PJP complications should be avoided in such cases.

## 1. Introduction

Except for idiopathic pulmonary fibrosis in the chronic phase, the use of steroids and immunosuppressive agents is often considered for treating other interstitial pneumonias [1]. Steroids are also often used in COVID-19 pneumonia to control the excessive inflammation associated with viral infections [2]. One of the most important factors to consider when using these drugs is the rise in infections associated with immunodeficiency [3]. *Pneumocystis jirovecii* pneumonia (PJP) is one such disease wherein immunosuppression is a risk factor and has a significant impact on prognosis. Therefore, patients with interstitial pneumonia (IP) taking steroids or immunosuppressive medications should be approached with caution. We present two cases of PJP at our institution, both of which occurred after COVID-19 infection in patients with IP on steroids.

## 2. Case Report

### 2.1. Case 1

A 69-year-old man diagnosed with IP in 2018 and receiving oral prednisolone (PSL) at a maintenance dose of 10 mg daily developed a fever in April 2020, and the polymerase chain reaction (PCR) test was positive for SARS-CoV2. He had not received the COVID-19 vaccine. Treatment began with oral favipiravir, which was widely used for COVID-19 treatment in Japan at the time. However, due to the patient’s lack of improvement and poor oxygenation, he was admitted to our hospital on the seventh day after his illness began. Upon examination, his body temperature was 36.2 °C, his heart rate was 51 beats/min, and his oxygen saturation was 94% (room air). The blood examination showed the following results: white blood cells (WBC) 30.1 × 10^3^/μL, hemoglobin (Hb) 14.8 g/dL, platelets 45.8 × 10^4^/μL, Na 136 mEq/L, K 4.2 mEq/L, Cl 97 mEq/L, C-reactive protein (CRP) 0.75 mg/dL, blood urea nitrogen (BUN) 23 mg/dL, creatinine (Cre) 0.88 mg/dL, aspartate aminotransferase (AST) 72 IU/L, alanine aminotransferase (ALT) 92 IU/L, and lactate dehydrogenase (LDH) 370 U/L. These findings are typical in the early stages of COVID-19 infection. Figure 1A depicts the patient’s clinical course. Ground-glass opacities and consolidation were seen on chest computed tomography (CT) (Figure 2). COVID-19 pneumonia was almost completely resolved after the fever subsided. The patient developed fever again on the 19th day after the onset of the disease (Day 19), and a chest CT scan revealed a new ground-glass opacity (GGO), thereby raising the possibility of pneumonia caused by a common bacterium. Levofloxacin treatment was ineffective, and an increase in serum β-D-glucan levels to 9.7 pg/mL increased the possibility of PJP. The fact that trimethoprim–sulfamethoxazole (TMP–SMZ) improved pneumonia led to a clinical diagnosis of PJP.

### 2.2. Case 2

A 70-year-old woman taking 4 mg of PSL orally daily for IP and rheumatoid arthritis was admitted to our hospital in April 2021 for COVID-19 treatment for 18 days (Days 7–25), where dexamethasone and tocilizumab were used. She had not received the COVID-19 vaccine. She began coughing one week after discharge (day 33), and five days later, she developed respiratory failure, with a chest CT revealing worsening pneumonia. Therefore, she was readmitted to the hospital (Day 38). During the second examination, her body temperature was 36.3 °C, her heart rate was 91 beats/min, and her oxygen saturation was 94% (2 L/min). The blood examination showed the following results: WBC 11.0 × 10^3^/μL, Hb 12.0 g/dL, platelets 14 × 10^4^/μL, Na 123 mEq/L, K 4.5 mEq/L, Cl 90 mEq/L, CRP 16.24 mg/dL, BUN 16 mg/dL, Cre 0.57 mg/dL, AST 23 IU/L, ALT 12 IU/L, and LDH 438 U/L. At this time, the quantitative SARS-CoV-2 antigen test was negative. Figure 1B depicts the patient’s clinical course. Her respiratory failure worsened after admission, her serum CRP was elevated, and a chest CT revealed an enlarged GGO (Figure 2). Ultimately, the diagnosis of PJP was made on the basis of high serum β-D-glucan levels and positive PCR for *P. jirovecii* DNA in the sputum. Thus, she was given TMP–SMZ treatment, but her respiratory failure worsened and she died on Day 49. She was subjected to a pathological autopsy. The lungs were clogged, and the histological assessment revealed multiple diffuse vitreous membranes, a sign of diffuse alveolar damage (DAD). Her collagen fiber increased in the interstitium, and numerous fibroblast foci were found in the alveolar wall and space. Furthermore, there was mucinous and exudate accumulation in the interstitium and alveolar space (Figure 3). Additionally, *P. jirovecii* cyst was identified. Although COVID-19 pneumonia and acute exacerbation of IP could cause DAD, PJP was determined to be the cause based on clinical and laboratory findings.

## 3. Discussion

The differential diagnosis of post-COVID-19 pneumonia should include COVID-19 pneumonia relapse, acute exacerbation of IP [4], secondary infectious pneumonia [5,6,7], and organizing pneumonia [8]. Here, PJP was diagnosed based on the CT findings, the elevated serum β-D-glucan levels, and a positive PCR test for *P. jiroveci* DNA in the sputum.

According to many reports, complications of bacterial infection and secondary infections are uncommon after COVID-19 pneumonia. Furthermore, there have been a few reports of PJP following COVID-19 infection [9], but there is currently no consensus. Some cytokines are known to affect not only the control of infections but also fibrosis of the lungs. Zhong JH et al. reported that lower circulating interferon-gamma is a risk factor for lung fibrosis in patients with COVID-19 infections [10]. There is also a report of cytokine-mediated involvement of tuberculosis and helminth infection in pulmonary fibrosis secondary to COVID-19 [11]. Although the involvement of cytokines in PJP after COVID-19 pneumonia with IP has not been investigated, there is room for further studies on this matter. Additionally, both cases in this study were non-vaccinated cases. A comparison of the risk of PJP in COVID-19 pneumonia between patients who are vaccinated and non-vaccinated is also the subject of a future study. Corticosteroids are recommended for treating COVID-19 pneumonia when respiratory failure due to the pathogenesis of acute respiratory disease syndrome (ARDS) becomes severe [2], but caution should be exercised because of the risk of developing PJP in some patients, as in these cases. During the two years from February 2020 to January 2022, 11 patients (1.1%) admitted to and treated at our hospital had underlying IP. Because we were unable to follow up with all of the patients, these data are only for reference purposes; however, the two cases shown here were those who developed PJP after their infection with COVID-19 was cured. Given that PJP was observed in two of the eleven patients with IP who had previously received steroid treatment, we believe that special attention should be paid to the development of PJP in patients with COVID-19 receiving steroid treatment for IP. There has been no pathological autopsy report of a case of PJP in a patient with IP after COVID-19 infection, and this is the first such case. Lesions primarily composed of DAD were found in the lungs. Along with the original IP lesions, the combination of ARDS-like effusion in the alveolar space caused by COVID-19 and an infectious lesion caused by PJP was the reason for the unfavorable outcome of this case. Although the number of patients presenting with pneumonia such as this case has decreased since Omicron strains became the mainstay of COVID-19 infection, we believe that these two cases highlight the importance of being cautious about secondary PJP development in the future.

## 4. Conclusions

In conclusion, using steroids and immunosuppressive medications increases the risk of developing PJP. However, since patients with IP who have been taking these medications for a long time frequently receive additional steroid treatment for COVID-19 pneumonia, we should be especially watchful for PJP complications in these situations.

## Figures and Tables

**Figure 1 medicina-58-01151-f001:**
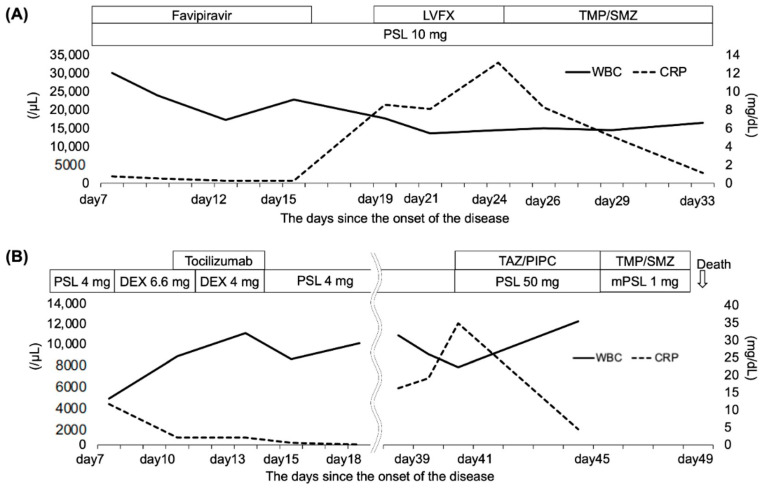
The clinical course of the patients. (**A**) case 1. (**B**) case 2. WBC: white blood cell, CPR: C-reactive protein, PSL: prednisolone, DEX: dexamethasone, LVFX: Levofloxacin, TMP/SMX: trimethoprim/sulfamethoxazole, TAZ/PIPC: tazobactam/piperacillin.

**Figure 2 medicina-58-01151-f002:**
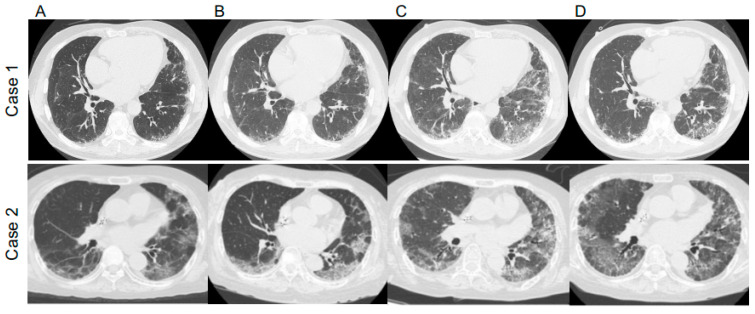
Chest computed tomography (CT) images of Case 1 and Case 2 (**A**) at COVID-19 diagnosis, (**B**) after treatment of COVID-19, (**C**) at *Pneumocystis jirovecii* pneumonia (PJP) diagnosis, and (**D**) after PJP treatment with trimethoprim–sulfamethoxazole.

**Figure 3 medicina-58-01151-f003:**
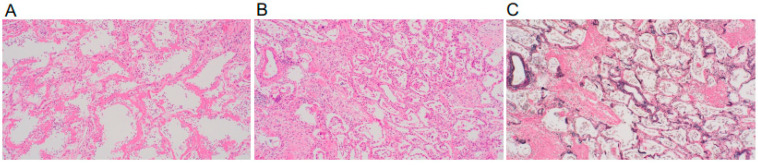
The lung sections obtained from autopsy were stained with the hematoxylin–eosin stain (**A**,**B**) and the Elastica van Gieson stain (**C**). Representative images of (**A**) hyaline membranes’ line alveolar spaces, (**B**) fibroblastic foci, and (**C**) fibrotic lung tissue are shown (original magnification × 200).

## Data Availability

Data supporting the study findings are available from the corresponding author upon reasonable request.

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
