# Peer review of "Pneumocystis jirovecii Pneumonia Associated with COVID-19 in Patients with Interstitial Pneumonia"

_medicina, 2022, doi:10.3390/medicina58091151_

Round 1

Reviewer 1 Report

- It has been a pleasure to revise the manuscript “Pneumocystis jirovecii pneumonia associated with COVID-19 2 in patients with interstitial pneumonia”, submitted for publication in Medicina. In general, I consider that we are in the presence of a useful and interesting work that describes two cases of patients with interstitial pneumonia (IP) on steroids treatment that develop Pneumocystis jirovecii pneumonia (PJP) following coronavirus disease 2019 (COVID-19).

 - Despite of the good description of the clinical and pathological characteristic of both patients, I consider that this document is unbalanced descriptive. That is, I believe that the authors do not adequately delve into the mechanisms by which steroid treatment can lead to the development of PJP after a COVID-19 event in a IP patient.

 - On the other hand, clinical, chest CT and post-mortem findings showed that fibrosis and restriction of pulmonary function are frequently present in patients who have recovered of COVID-19 (Fonte L, et al. Overlapping of Pulmonary Fibrosis of Postacute COVID-19 Syndrome and Tuberculosis in the Helminth Coinfection Setting in Sub-Saharan Africa. Trop Med Infect Dis. 2022; 7:157. doi.org/10.3390/tropicalmed7080157). In fact, PF is already recognized among the most important sequelae of SARS-CoV-2 infection (Hu ZJ, et al. Lower Circulating Interferon-Gamma Is a Risk Factor for Lung Fibrosis in COVID-19 Patients. Front Immunol (2020) 11:585647. doi: 10.3389/fimmu.2020.585647). I suggest the authors consider in the discussion of their work the possible participation of post-Covid fibrosis in the fatal evolution of their patients.

 - I general, I considered that this interesting review must offer more authors opinion in relation the data they describe.

Author Response

Response: Thank you for this suggestion. As pointed out, we cited these papers and discussed in the Discussion section.

Reviewer 2 Report

The authors have made an interesting attempt on “Pneumocystis jirovecii pneumonia associated with COVID-19 2 in patients with interstitial pneumonia.” The manuscript is interesting; however, the authors need to justify the scientific writing manuscript. Some of the general comments are provided below:

1.     Conclusion should be included. in the abstract.

2.     What was the COVID-19 vaccine status of both cases?

3.     It would be interesting if the authors had compared the risk of pneumonia between vaccinated and non-vaccinated individuals. 

Author Response

Response: We appreciate the positive feedback and helpful suggestion. We revised the abstruct. Also, a review of the medical records revealed that both of the two cases in this study were non-vaccinated cases. The risk of pneumonia between vaccinated and non-vaccinated individuals seems to be a very interesting research question, but since no conclusions can be drawn from these two cases, we would like to make it an issue for future study. We discussed this in the discussion section.

Reviewer 3 Report

The authors focused on the study of Pneumocystis jirovecii pneumonia associated with COVID-19 in patients with interstitial pneumonia. This is an interesting and comprehensive Case Report. The article is well structured.

In my opinion:

- The abstract presents an accurate description of this study.

- An Authors was conducted adequate literature review

- References support the rationale for reporting the study.

- Patients are described adequately.

- The management of the study is effectively described.

- Valid and reliable outcome measures are utilized.

- Conclusions are appropriate.

Key points to consider:

Line 49 and 76 - Cl instead CL - please correct

Please edit Figure 1, as it is poorly readable in this form

Financing: - please complete

Informed Consent Statement: - please complete

The distance between line 141 and 142 is too large, please correct this

Overall impression about the quality of the study is very good.

Author Response

Response: Thank you for this suggestion. As pointed out, we have revised these points.

Round 2

Reviewer 1 Report

Dear authors,

You refer to only one of the three comments/suggestions I made to you in the first review. No problem, they were made with the intention of improving your work

You said “As pointed out, we cited these papers and discussed in the Discussion section”. Certainly, you cited the paper mentions in one of our comments, which was optional. However, the text of the sentence that you use to refer to those papers is, at least, inexactly. In neither of these two articles is it expressed “However, some reports indicated that cytokines such as interferon-γ are involved in pulmonary fibrosis and …10-11”. This error must be corrected, because the authors of those two works expressed different contents. The title of one of those papers (“Lower Circulating Interferon-Gamma Is a Risk Factor for Lung Fibrosis in COVID-19 Patients”) is already different from what was written by the authors.

Best regards,

Author Response

Thank you for your additional comments. We changed the expressions on the discussion. Following your advise, reference #10 and #11 were explained separately.